# The Implications of Sports Biomechanics Studies on the Research and Development of Running Shoes: A Systematic Review

**DOI:** 10.3390/bioengineering9100497

**Published:** 2022-09-22

**Authors:** Shuangshuang Lin, Yang Song, Xuanzhen Cen, Kovács Bálint, Gusztáv Fekete, Dong Sun

**Affiliations:** 1Faculty of Sports Science, Ningbo University, Ningbo 315211, China; 2Doctoral School on Safety and Security Sciences, Óbuda University, 1034 Budapest, Hungary; 3Savaria Institute of Technology, Faculty of Informatics, Eötvös Loránd University, 9700 Szombathely, Hungary

**Keywords:** running shoes, biomechanics, performance, injuries, development

## Abstract

Although various sports footwear demonstrated marked changes in running biomechanical variables, few studies have yielded definitive findings on the underlying mechanisms of shoe constructions affecting running-related performance and injuries. Therefore, this study focused on examining the effect of basic shoe constructions on running biomechanics and assessing the current state of sports shoe production in terms of injury and efficiency. Relevant literature was searched on five databases using Boolean logic operation and then screened by eligibility criteria. A total of 1260 related articles were retrieved in this review, and 41 articles that met the requirements were finally included, mainly covering the influence of midsole, longitudinal bending stiffness, heel-toe drop, shoe mass, heel flare, and heel stabilizer on running-related performance and injuries. The results of this review study were: (1) The functional positioning of running shoe design and the target groups tend to influence running performance and injury risk; (2) Thickness of 15–20 mm, hardness of Asker C50-C55 of the midsole, the design of the medial or lateral heel flares of 15°, the curved carbon plate, and the 3D printed heel cup may be beneficial to optimize performance and reduce running-related injuries; (3) The update of research and development concepts in sports biomechanics may further contribute to the development of running shoes; (4) Footwear design and optimization should also consider the influences of runners’ strike patterns.

## 1. Introduction

Footwear can be a highly effective instrument for manipulating human movement [1,2]. By promoting core technology and refining material property, major shoe companies have been marked as high-tech bellwethers of the global athletic footwear industries [1]. In the past 40 years of rapid development, many sports shoe manufacturers have tried to incorporate specific functions into their prototypes, including cushioning, stability, energy return, and motion control [1,2]. At present, scientific research has paid considerable attention to the practicality of footwear, inspired by numerous assertions of running shoe companies when marketing products that are intended to optimize performance or block sports injuries. Despite decades of efforts in sports shoe design, the prevalence of lower limb injuries related to running has not seen a dramatic decline [3,4].

Running loads ranging from 1.5 to 3 times the body weight are repeatedly consumed by each leg. Such continuous loading and subsequent force shocks may inflict microtrauma and muscle fatigue, resulting in impaired function [5]. Given that repeated loading of the musculoskeletal system is often regarded as a predictor of damage occurrence, shoes with softer, more shock-absorbing soles have been highlighted as tactics responsible for mitigating the ground reaction force [6,7]. Nigg [3], in contrast, proposed that impact forces may not be a major factor in the development of injuries. According to relevant epidemiological data, Hao [4] verified that modern running shoe technology appears ineffective in reducing runners’ lower limb and foot injuries. In addition, among the most striking patterns arising from studies of sports shoe biomechanics, many results are the indirect product of shoe-induced kinematic changes. Since a comprehensive upgrade in athletic footwear is warranted, recognizing biomechanical adaptations aligned with shoe technologies is vital to understanding the mechanism of any potential outcomes.

As one of the most notable footwear elements, the sole has undergone massive adjustments (e.g., sole geometry and properties). In the previous articles, the role of sole shoe characteristics on running biomechanics was discussed. Chambon [8] stated that increased midsole thickness had no effect on foot-strike patterns or kinematics but influenced stance phase duration. A broad range of heel-to-toe drops applied in running shoes has been found to stimulate distinct foot-strike patterns and injury potentials among different running crowds (i.e., regular, occasional, or untrained) [9]. Moreover, Willwacher et al. [10] deduced that variable stiffness shoes could increase anterior ground reaction force, which supports athletes’ forward propulsion and potentially improves running efficiency. By offloading foot joints during locomotion, an enhanced forefoot bending stiffness can serve a preventative role in running injuries [11]. On the other hand, Law et al. [12] concluded that individuals might respond inconsistently to the modification of sole stiffness owing to the disparity of gait and muscle activation patterns.

Although there are also various sports shoe constructions that have demonstrated marked changes in running biomechanical variables, such as shoe mass, heel flare, and heel stabilizer [1,4], few studies have yielded definitive findings on the mechanisms underlying shoe features affecting running-related performance and injuries. To underpin modern sports shoe designs, this study focused on examining the effect of differing sole features on running biomechanics and assessing the current state of the production of sports shoes in terms of injury and efficiency.

## 2. Materials and Methods

### 2.1. Protocol Registration

This review was conducted in accordance with the Preferred Reporting Items for Systematic Reviews and Meta-Analyses (PRISMA) statement (INPLASY202280107).

### 2.2. Search Strategy

The following keyword combinations were used in a standardized electronic literature search process: “running shoes” OR “running footwear” AND (“midsole” OR “midsole stack height” OR “cushioning” OR “stiffness” OR “bending stiffness” OR “heel-to-toe drop” OR “shoe weight” OR “heel flare” OR “heel cup” OR “heel counter”) from 1 January 1980 to 1 June 2021, via the five electronic literature databases (Google Scholar, PubMed, ScienceDirect, Scopus, and Web of Science). A snowballing procedure was conducted to review the reference list and identify new papers. The search and selection processes are summarized in Figure 1.

### 2.3. Eligibility Criteria

Eligibility criteria of the literature in this study: (1) Original research from peer-reviewed English journals was included. Conference papers, review papers, master (doctoral) graduation papers, case studies, and non-full text articles were excluded. (2) The research must be related to the midsole (hardness, thickness, and material), bending stiffness, heel-to-toe drop, shoe mass, heel flare, and heel stabilizer (heel cup, heel counter) of running shoes. (3) The research must involve the corresponding statistical analysis and offer quantitative results on the influences of shoe construction in biomechanical changes during running that are associated with performance and/or running injuries; non-biomechanical related studies were excluded.

### 2.4. Data Extraction and Management

All the articles in this study were downloaded by the same author and imported into Mendeley Reference Management Software (Elsevier Ltd., Amsterdam, The Netherlands) for duplicate elimination, articles management, and citation. The other two authors conducted data extraction and analysis of the literature, mainly including the first author’s name, research publication year, country, research design, athletic performance-related and/or injuries-related biomechanics variables, and the primary results. Disagreements regarding data extraction were resolved by the corresponding authors if happened. The PRISMA checklist was followed to synthesize data.

### 2.5. Quality Assessment

Cochrane Risk of Bias Assessment Tool was used to assess the risk of bias in each study by two authors. Disagreements of quality assessment were resolved by the corresponding authors if happened. Seven domains were evaluated (random sequence generation, allocation concealment, blinding of participants and personnel, blinding of outcome assessment, incomplete outcome data, selective reporting, and other biases) and each domain has three grades, i.e., low risk of bias, unclear risk of bias, and high risk of bias.

## 3. Results

### 3.1. Basic Characteristics of Included Studies

The comprehensive research generated 1230 articles (as shown in Figure 1). After removing duplicated articles, a total of 596 articles was included. After two authors conducted a title and abstract screening, a total of 125 articles was included. The articles were further screened through the three eligibility criteria, and 41 related articles were finally used in this study.

According to the authors’ country/region, 17 studies were from Europe, including 4 from France, 3 from Germany, 3 from Luxembourg, 2 from the United Kingdom, 2 from Austria, 1 from Belgium, 1 from Spain, and 1 from Switzerland; a total of 19 studies were from North America, including 10 from Canada and 9 from the United States, and 6 from Asia, including 5 from China and 1 from South Korea. The proportion of research literature published in 1980–1999, 2000–2009, and 2010–2021 was 11.90%, 23.81%, and 64.29%, respectively, which clearly shows that the interest in this area is increasing. The effects of various shoe characteristics on running performance and/or injury will be further discussed below.

### 3.2. Risk of Bias

The risk of bias in the 41 studies was assessed, and the consensus was reached after discussion. The overall results are shown in Figure 2, and 64% of the studies reported participants’ randomization. A total of 47% of the studies reported the method of allocation concealment (subject random assignment), 17% of the studies did not fully describe it, and the rest did not include allocation concealment. Only 5% of the studies were double-blind, 83% were not fully reported, and only one study made it clear that double-blind was not used. A total of 74% of the studies described the blind method used in the evaluator, which was not fully reported in other studies. Only one study did not fully report whether the data were missing. All the studies recorded their research plan and researched according to the program.

### 3.3. Effects of the Midsole on Running Performance/Injury

The main design aspects of running shoes are shown in Figure 3. A total of 12 articles (Table 1) investigated the impact of midsole hardness designs on running performance/injury [5,6,9,13,14,15,16,17,18,19,20,21]. A total of two articles investigated the effect of midsole thickness designs on running performance/injury [8,12]. A total of three articles investigated the impact of midsole designs (materials and structures) on running performance/sports injury [22,23,24].

#### 3.3.1. Midsole Hardness

In the running, the increase in midsole hardness (i.e., from Asker C40 to Asker C65) was linked to sports performance as specified by less energy expended at the metatarsophalangeal and less peak rearfoot eversion velocity [13,20]. In contrast, other studies found that increased midsole hardness of running shoes had no significant effect on peak tibial acceleration, running speed, walking time, and lower limb muscle activity (medial femoris, biceps femoris, and gastrocnemius) [5,16]. In terms of running-related injuries, findings remain inconclusive. Contradicting the general concept that softer midsole shoes can prevent damage in running by dispersing impact mass, Baltich et al. [20] claimed that the peak value of the vertical ground reaction force and the stiffness of the knee and ankle joints were found reduced as the midsole hardness of running shoes increased. Similarly, after tracking 247 runners for 5 months, Theisen et al. [18] deduced that the injury incidence associated with applying a softer midsole might be higher than using a more rigid midsole. However, the present finding indicated that regardless of the injury’s location, type, or severity, there are no statistically significant differences between running shoes with varying midsole hardness.

The other three studies emphasized the impact of hardness in different midsole areas on running performance/sports injuries. Sterzing et al. [6] evaluated the effect of running shoes with varying midsole hardness on the rearfoot and forefoot. Findings suggested that the cushioning properties of the rearfoot almost entirely determine the vertical ground reaction forces and lower extremity kinematics. Moreover, running shoes that combine a soft forefoot and a stiff rearfoot midsole can effectively improve the cushioning characteristics. By examining the effects of the medial and lateral stiffness of the midsole on lower limb running biomechanics, Dixon et al. [21] found that the hindfoot valgus angle and peak loading rate of ground reaction force were significantly increased when running in the lateral hardness shoes. Oriol et al. [17] evaluated the effect of randomly varying medial dual-density midsole elements in the running. They concluded that although the midsole hardness varies from the length and position, there is no significant difference in the biomechanical parameters, i.e., vertical impact force peak during running.

#### 3.3.2. Midsole Thickness

Contrary to the previous hypothesis, different running shoe midsole thicknesses did not change the running patterns because of the impact of relevant biomechanical parameters (e.g., landing angle, ground reaction force). Chambon et al. [8] found that greater midsole thickness has little impact on kinematics, force, and acceleration variables, but it will significantly enhance the stance-phase duration. In line with this view, Law et al. [12] reported that increased midsole thickness might reduce the vertical loading rates and growth in the contact time. Moreover, it appears that footwear with varying midsole thickness has little effect on foot strike pattern, stride duration, or cadence.

#### 3.3.3. Midsole Material and Structure

Ethylene-vinyl acetate copolymer (EVA) and polyurethane (PU) are currently widely used in midsole materials of running shoes [24]. Different materials have varying cushioning and resilience properties, which may affect running efficiency and injuries. Wang et al. [22] compared the sports performance of midsole running shoes made of EVA and PU, respectively. The results showed that PU midsole running shoes lead to higher durability, while EVA midsole running shoes reinforced energy returns more than PU ones in a running distance of 0–500 km.

In addition to EVA and PU materials, more attention has been paid to the leaf spring-structured midsole shoe (which replaces the traditional midsole construction by leaf spring) in recent years. By comparing the difference between leaf spring-structured midsole shoe (LEAF) and standard foam shoe (FOAM) on Spatio-temporal variables and running economy, Wunsch et al. [24] found that the midsole running shoes with leaf spring structure can significantly increase the step length, which accounts most for an improved running economy. In the same year, the effect of a LEAF on joint mechanics and lower limb muscle forces during overground running was contrasted to a FOAM. Wunsch et al. [23] stated that LEAF could improve running performance by substantially reducing energy absorption at the hip joint and energy production at the ankle joint.

### 3.4. Effects of the Bending Stiffness on Running Performance/Injury

Seven articles (Table 2) examined the impact of bending stiffness of running shoes on running performance and related damages [13,19,25,26,27,28,29].

The adjustment of the bending stiffness of running shoes was associated with running performance and running economy. According to Hoogkamer et al. [29], by applying shoes with increased bending stiffness, all 18 tested participants have decreased the energetic cost of running. Similarly, Roy et al. [26] clarified that by increasing the bending stiffness of running shoes, the running economy is significantly improved (1% energy saving). However, there was no significant difference in the energy absorbed by metatarsophalangeal (MTP) joints and activities of muscles such as soleus, gastrocnemius, and rectus femoris in running. Stefanyshyn et al. [13] found that the increase of bending stiffness reduces the energy loss of the MTP joint and improves the performance of the MTP joint during running and vertical jumping (improved vertical jump height). A similar view is held by Willwacher et al. [19], who demonstrated that as the bending stiffness of a running shoe increases, the negative work of the MTP joints decreases. In contrast, the positive result increases significantly during running. By analyzing the influence of different bending stiffness of running shoes on sprint performance, Stefanyshyn et al. [25] concluded that speed efficiency could be improved by increasing the shoe bending stiffness.

Other researchers hold an opposite standpoint towards bending stiffness of running shoes on the running economy. As Madden et al. [27] proposed, increasing the forefoot bending stiffness of the running shoes did not significantly affect the overall running economy. Moreover, they believed that improved running economy with increased forefoot bending stiffness is not due to a decrease in negative work at the MTP joint. There are two main influences on the foot caused by the relative increase in the bending stiffness of running shoes: (1) the metatarsal-phalangeal joint’s bending was diminished (i.e., less mechanical energy was lost at the joint); (2) the lever between the resulting ground reaction force and the ankle joint was increased, allowing for the production of more extraordinary ankle joint moments if the triceps surae was sufficiently robust [30]. Nigg et al. [30] further stated that the increase in the ankle joint moment was the dominant effect brought about by the change in the bending stiffness of running shoes. The increase of ankle torque was the dominant effect caused by the bending stiffness of running shoes. This theory has been extended further by Roy et al. [26], who supported that the increase in bending stiffness of running shoes leads to a raised peak moment of the ankle joint during running.

Furthermore, the two articles implied that increases in the bending stiffness of running shoes result in substantial increases in running propulsion and stance duration [19,28], which is consistent with the previous findings of Nigg et al. [30], who indicated that increasing the ground contact time and the propulsive force by altering the bending stiffness of running shoes could effectively reduce the energy loss, thereby promoting the running economy.

### 3.5. Effects of Heel-to-Toe Drop on Running Performance/Injury

Seven articles (Table 3) discuss the impact of the heel-to-toe decline of running shoes on running performance and injury [7,11,31,32,33,34,35].

As a brand-new “barefoot” running shoe, minimalist shoes have favored the public and scientific researchers in the last decade. A minimalist shoe with a lower drop might result in a biomechanical change toward a forefoot strike pattern. According to Chambon et al. [11], the 0 mm drop shoe version generated a striking design comparable to a forefoot strike at touchdown in the 8 mm drop shoe version. Moreover, an opposite ground reaction was found on wearing running shoes with different shoe drops on overground and treadmill running, which may be caused by the kinematic changes at the moment of landing.

It has been suggested that a higher shoe drop may increase knee abduction at the mid-stance phase, influencing sagittal plane and flexion angle, and decrease tibial acceleration, metatarsal, and knee extension angle in landing [31,32,33]. By investigating the influences of shoe drop on running mechanics, Besson et al. [35] indicated that larger heel-to-toe drop conditions would increase net knee flexion moment (*p* < 0.001) in the push-off phase, but also decrease net joint ankle flexion moment during the braking phase (*p* < 0.001). Thus, a more significant drop may benefit women with a stiff Achilles tendon, such as high-heeled shoe wearers, while a shoe with no drop can be an excellent option for women who suffer from knee pain or fatigue.

The other two articles included a long-term follow-up survey on the effects of different shoe drops on sports performance and sports injuries. Among 553 participants examined, Malisoux et al. [7] found no significant relationship between the overall risk of sports injury and the difference between shoe drops of running shoes. However, low drop running shoes were found to reduce the injury rate of occasional runners, but it seems risky for regular runners. In the following year, Malisoux et al. [34] investigated the long-term consequences of wearing varying height heel-to-toe drop shoes in the following year. The results showed that apart from knee abduction during the mid-stance phase, no discrepancies in spatiotemporal variables or kinematics were observed between shoe versions of varying drop heights during this 6-month follow-up.

### 3.6. Effects of Shoe Mass on Running Performance/Injury

Three articles (Table 4) examined the impact of running shoe mass on running-related performance and damages [36,37,38].

While determining whether shoe mass accounts for the increased oxygen consumption associated with shod running or barefoot running, Divert et al. [36] proposed that the increased metabolic cost associated with shoe running was attributable to the added mass on the shoe. This idea has been developed further by Franz et al. [37]. They reported that submaximal oxygen uptake (VO2) increases by approximately 1% for each 100 g applied per foot, although there were no significant differences in VO2 or metabolic power between barefoot and shod running. Similarly, the metabolic rate grew about 1.11% per 100 g per shoe by adding mass to the shoes. In addition, each additional 100 g per shoe decreases running economy and proportionately slows 3000 m time trials results [38].

### 3.7. Effects of Heel Flare on Running Performance/Injury

Three articles (Table 5) included the impact of lateral heel flare (flared heel) on running performance and related injuries [39,40,41]. By analyzing the influence of different heel flares on rearfoot movement during running, Clarke et al. [39] found that shoes with 0° heel flare made for considerably more maximum pronation and total rearfoot motion than shoes with 15° or 30° heel flares. Moreover, there were few noticeable variations between the 15° and 30° flare conditions. Nigg et al. [40] further studied the association between lateral heel flare, impact forces, and pronation. Results revealed that increasing the heel flare improves initial pronation, and it has no effect on the magnitude of total pronation or the importance of the impact force peaks. Additionally, the study of Stacoff et al. [41] aimed at quantifying the effects of lateral heel flares on the stance phase of running. Findings showed that altering the lateral heel flares had no impact on talocalcaneal rotations. Moreover, a possible relationship or coupling effect was found between the heel flares and bone eversion.

### 3.8. Effects of Heel Stabilizer on Running Performance/Injury

Five articles (Table 6) explored the impact of heel cups on running performance and related damages. Among them, two focused on the effects of running shoes heel cup on running performance/sports injury [42,43], and three focused on the impact of running shoes heel stabilizers on running performance and related injuries [44,45,46].

Findings of Li et al. [43] showed that after four weeks of wearing the individualized heel cup, it was found to help secure the skeletal system and soft tissue of the plantar heel when walking and jogging and significantly minimize self-reported discomfort. Furthermore, the cushioning effect of heel cups has been proved by Wang et al.’s [42] study. As results suggested, rubber and plastic heel cups have achieved their cushioning effect through various mechanisms. Therefore, the heel cup selection process should be driven by the pathogenesis of the heel pain, i.e., rubber heel cups were suggested for inflammation patients; plastic heel cups were indicated for heel pad atrophy patients.

According to Alcantara et al. [46], shoe markers greatly underestimated calcaneus range of motion (ROM) across all planes of action, representing inadequately reflecting calcaneus motion. Moreover, there were no improvements in tibial transverse plane ROM following heel counter modifications, implying that any changes in heel counter rigidity produced by the amendments did not affect tibial rotation. By examining the effects of upper vamp components on pronation and torsion of the foot, Ferrandis et al. [45] found that prototype 2 (with an external heel counter) and prototype 4 all have a lower rearfoot angular difference (with a rear lace anchor from the last eyelet of the lacing to the midsole embracing the tarsus). However, the rearfoot angles closest to the ground correspond to prototype 3 (with a rear lace anchor from the last eyelet of the lacing to the midsole embracing the tarsus). Jorgensen [44] examined the effect on muscle load of the cumulative effect of increased shock absorption and stabilization provided by the heel counter. The results indicated that a properly installed rigid heel counter in the shoe resulted in a 2.4% substantial decrease in VO2, decreased musculoskeletal transients, and decreased triceps surae and quadriceps muscle function at heel attack, both of which increase running economy.

## 4. Discussion

The purpose of this study was to outline the impact of various sole features on running biomechanics associated with performance and injury risk.

### 4.1. Midsole Properties

Given existing evidence, the influence of midsole hardness on running-related performance and injury remains controversial. The current study results show that running shoes with a reasonable range of midsole hardness (i.e., 50–52 Asker C) seem more feasible since it mitigates injury risk and improves running efficiency. Moreover, the midsole hardness can be distinguished regarding individual differences between posterior-anterior and mediolateral segmented cushions to improve perceived comfort [6,21].

Shoes with thicker midsole were more effective in mitigating impact because it allows more material deformation; in this case, they can also prolong the stance phase of running, which facilitates energy storage and return [8,12]. However, the findings of how much energy can be stored and produced during support periods of 6–13 months have not been quantified, and the appropriate time and frequency of energy return on the take-off moment remain inconclusive [47,48,49]. This study, therefore, implied that medium-thickness running shoes (e.g., 15–20 mm) seem more credible for the present application.

From the existing research, EVA midsole material represents the lighter weight and excellent energy return ability, optimizing performance while helping prevent injuries [22,50]. What is more, the blends of EVA and other polymer materials, such as PHYLON, also known as secondary foaming materials, are more effective than EVA materials in damping performance, elasticity, and durability. In addition, in recent decades, BASF has jointly launched new midsole materials, E-TPU (foamed TPU) and Elastopan^®^ Sports Light, in cooperation with top international sports shoe manufacturers (e.g., Adidas (Herzogenaurach, Germany) and Brooks, (Seattle, WA, USA)), to improve the cushioning capacity and durability. Since there is a significant price difference in the midsole materials mentioned above, it is necessary to select appropriate midsole materials regarding the actual functionality of running shoes [23,24].

### 4.2. Longitudinal Bending Stiffness

The optimization of the bending stiffness of running shoes is inseparable from the individual runner characteristics. It not only depends on the runner’s weight, ankle joint extensor strength, etc., but is also closely related to the running speed, the ground reaction force, and the lever arm length of the lower limb joints [10].

A “U-shaped” curve relationship implied an ideal longitudinal bending stiffness for improving the running economy. For example, running shoes with 38 N/mm bending stiffness might have higher overall benefits than running shoes with 18 N/mm and 45 N/mm bending stiffness [26]. Moreover, the “teeter-totter effect” proposed by Nike Vaporfly may help renew the bending stiffness design of running shoes. Compared to the previous prototype shoe with curved carbon fiber plates, Nike Vaporfly used a stiff curved plate, which can increase the force given to the foot’s heel at take-off and gain performance up to 4–6% over standard footwear [30].

### 4.3. Heel-To-Toe Drop

The findings of the impact of the heel-to-toe drop on the landing patterns of running remain inconclusive. However, from the existing studies, the design of heel-to-toe drops of running shoes needs to consider the positioning of running shoes and the target group. Malisoux et al. [7] suggested that a slight decrease of heel-to-toe drop (e.g., 0–4 mm) is more feasible for the general population, supporting sports efficiency and keeping runners’ safety. Meanwhile a higher heel (e.g., 6–10 mm) should be considered in the design of professional competitive running shoes, or it may be customized according to the runner’s habits and biomechanical parameter changes in running. In addition, it is worth noting that there is also a running shoe demand difference for runners with different foot strike pattern for heel-to-toe drop since hindfoot running relies more on a positive drop while forefoot running relies on a negative one. More studies concerning this point are warranted for further clarification.

### 4.4. Shoe Mass

The increase in shoe mass will affect the perceived comfort and significantly affect the ankle joint angle, force moment, and plantar pressure during running [51]. Moreover, the increase in the shoe mass can increase energy consumption and reduce the economy of running [36,37,38]. Therefore, this study suggests that the shoe mass should be minimized after running shoes meet other design demands, or apply new materials with low weight and consistent ability to reduce the overall shoe mass.

### 4.5. Heel Flare

The increase of the heel flare at the medial side can reduce the rearfoot movement during running, especially excessive foot pronation, which has been described as a significant indicator of ankle joint injuries [39]. However, there is no clear conclusion about the relationships between running performance/injuries and different heel flares. This study proposes that the heel flare should adopt the standard angle of about 15°, which might initially increase certain foot pronation, but it had little effect on the peak value of vertical impact force [40,41]. In addition, the increase of the lateral heel flare can provide more support for the rearfoot at the moment of running landing, thus reducing the vertical impact force per unit area. Furthermore, such an increase in heel flare can reduce the ankle joint sprain risk by promoting the outward movement of the contact point between the ground and the shoe, increasing the length of the moment arm, and the rapid internal rotation of the subtalar joint at the landing moment [52].

### 4.6. Heel Stabilizer

The clinical significance of a personalized heel cup made by combining 3D scanning and 3D printing for plantar heel pain has been proven. Accordingly, it can serve as a treatment or intervention for foot disorders [43]. However, the material of the heel cup must be personalized regarding the patient’s conditions to find out the most significant benefits [42]. Furthermore, the perfectness of heel cup design among the public or even professional sports running shoes needs to be further verified.

The heel stabilizer of running shoes is mainly used to strengthen the function of the heel cup and control the stability of the rearfoot during running. As the heel cup cooperates, the heel stabilizer can effectively minimize rearfoot eversion and torsion, thereby reducing the risk of potential running injuries. For different groups of people/patients, the use of a heel stabilizer + vertical attachment or further combination with heel tightening can fully ensure the stability of the hindfoot during running [45,53,54].

## 5. Conclusions

In summary, most of the studies have focused on investigating the impact of running shoe midsoles, bending stiffness, and heel-to-toe drop on running performance and injuries, while few studies on running shoe mass, heel flare, and heel stabilizer have been established. Existing studies have initially found the impact of these structural parts on running economy and stability, and it was found that thickness of 15–20 mm, hardness of Asker C50-C55 of the midsole, the design of the medial or lateral heel flares of 15°, the curved carbon plate, and the 3D printed heel cup may be beneficial to optimize performance and reduce running-related injuries. Nevertheless, it is valuable to conduct more examinations regarding these exclusive features to enhance the credibility of research results and offer additional insights into running shoe designs. Overall suggestions for future studies are as follows: (1) More attention on the long-term effects of running shoe constructions on running and the underlying biomechanical mechanism of running-related injuries. (2) Concerning the specificity, runners’ basic information should be collected (e.g., anthropometric parameters, foot morphology, and running experience) for footwear design; (3) The update of research and development concepts in sports biomechanics (e.g., “teeter-totter effect” of curved carbon fiber plate) may further contribute to the development of running shoes; (4) Footwear design and optimization should also consider the influences of runners’ strike patterns.

## Figures and Tables

**Figure 1 bioengineering-09-00497-f001:**
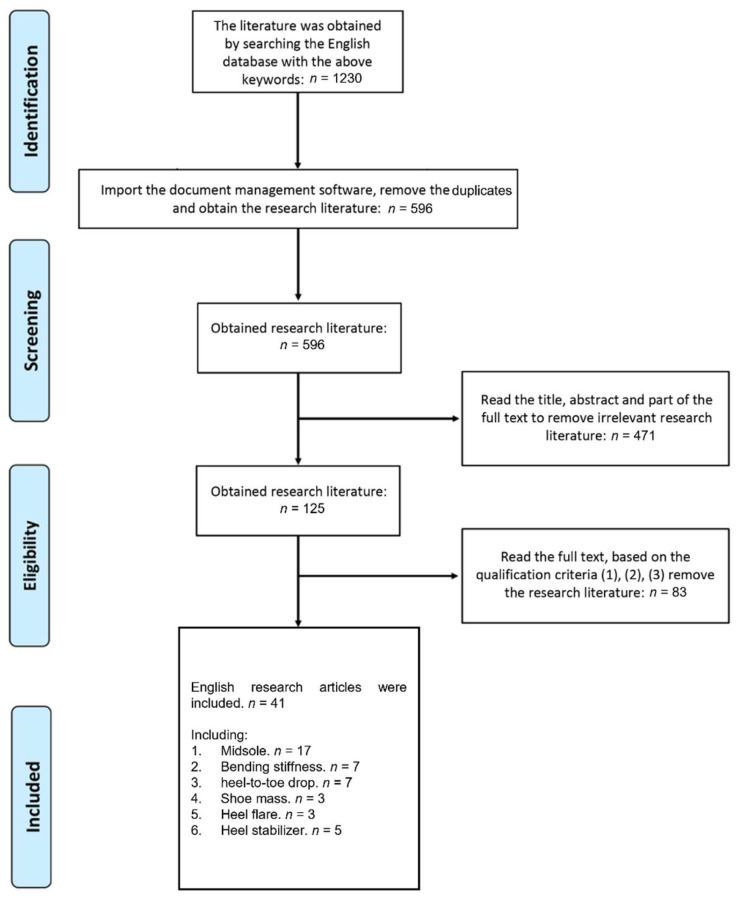
PRISMA Flow Chart for Systematic review.

**Figure 2 bioengineering-09-00497-f002:**
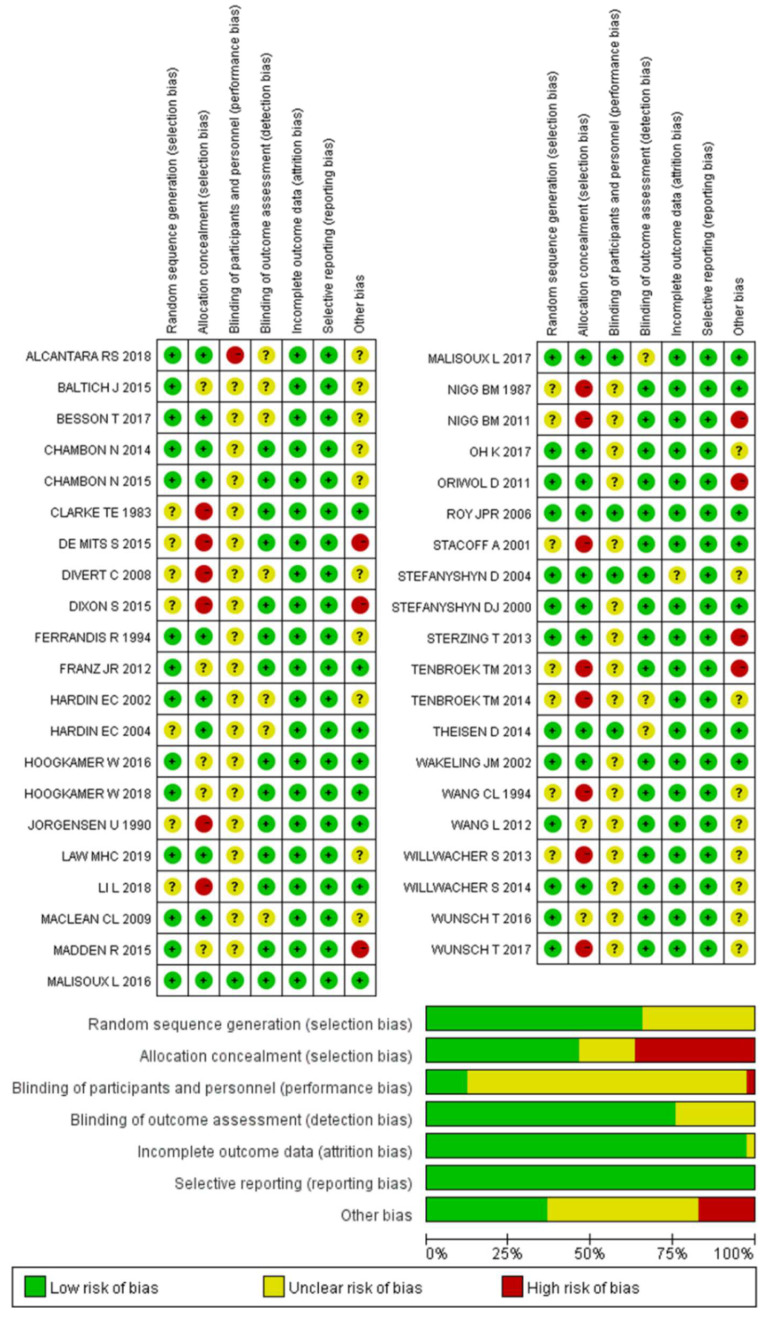
The result of the risk of bias assessment.

**Figure 3 bioengineering-09-00497-f003:**
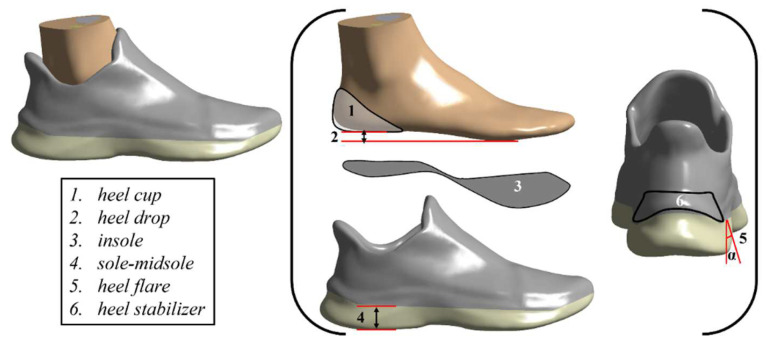
The main design aspects of running shoes.

**Table 1 bioengineering-09-00497-t001:** The influence of running shoes’ midsole on running performance and related injuries.

Author and Year	Country	Experimental Shoes	Participants	Methods	Results	
					Performance	Injury
1. The influence of midsole hardness on running performance/injury						
Baltich et al. [20]	Canada	Shaw C40 (soft) Shaw C52 (medium) Shaw C65 (hard)	93 Male 47, Female 46 16–75 years old	Running speed: 3.33 ± 0.15 m/s; Heel strike; Ground running	Soft ankle joint stiffness ↑ (medium-hard) Soft knee joint stiffness ↑ (female, medium-hard) Stiffness of the soft knee joint ↑ (male, medium)	The peak value of soft vertical ground reaction force (medium-hard)
Dixon et al. [21]	Britain	Shaw C 52 (CON) Shaw C52 + lateral midsole Shaw C60 (LAT1) Shaw C52 + midsole and outsole	10 Female over 50 years old	Running speed: 3 m/s; Ground running NA	LAT1 knee adduction angle ↓ (CON)	There is no difference between knee abduction peak torque and hindfoot valgus angle peak LAT2 ground reaction force loading rate and hindfoot valgus angle (CON)
Hardin et al. [5]	America	Shaw A 40 (soft) Shaw A55 (medium) Shaw A70 (hard)	24 Male NA	Running speed: 3.4 m/s; Treadmill downhill running; Heel strike	There is no difference in peak acceleration of the tibia	NA
Hardin et al. [14]	America	Shaw A 40 (soft) Shaw A70 (hard)	12 Male NA	Running speed: 3.4 m/s; Treadmill running; Heel strike	Peak velocity of dorsiflexion of stiff ankle ↑ (soft)	NA
Maclean et al. [15]	America	Shaw C40 (soft) Shaw C55 (medium) Shaw C70 (hard)	12 Female 19–35 years old	Running speed: 4.0 ± 5% m/s	Peak velocity of valgus of the hard hindfoot ↓	NA
Nigg et al. [16]	Canada	Shaw C40 (soft) Shaw C52 (medium) Shaw C65 (hard)	54 Male 36, Female 18 33.9 ± 20.1 years old	Running speed: 3.33 ± 0.17 m/s	There was no difference in muscle activity (frequency and time) of the biceps femoris muscle and gastrocnemius muscle of the lower extremities	NA
Stefanyshyn et al. [13]	Canada	General midsole Normal midsole + 3 mm carbon fiberboard Normal midsole + 5 mm carbon fiberboard	5 Male 32.0 ± 13.8 years old	Running speed: 4.0 ± 0.4 m/s; Ground running	The harder the metatarsophalangeal joint, the more energy loss ↓ There was no difference in energy production and absorption of the hip, knee, and ankle of the lower extremities There is no difference between energy storage and reuse of the metatarsophalangeal joint	NA
Sterzing et al. [6]	German	Midsole 50 ± 1 Asker C Soft-RF/soft-FF (SS) Medium-RF/medium-FF (MM) Hard-RF/hard-FF (HH) Soft-RF/hard-FF (SH) Hard-RF/soft-FF (HS)	28 Male 23.8 ± 2.0 years old	Running speed: 3.3 ± 0.1 m/s; Ground running; Heel strike	The softer the maximum metatarsal flexion and the internal rotation speed ↓ (the harder) MM sagittal plane landing angle ↓ (SH&HS) There is no difference in stance time	SH, SS, & MM first peak loading rate (HH & HS) SH second peak loading rate “(MM, HH, & HS) SS second peak loading rate” (HH & HS) MM second peak loading rate ↓ (HH)
Theisen et al. [18]	Luxembourg	Shaw C, 57.02 ± 2.96 (soft) Shaw C, 64.47 ± 2.22 (hard)	247 male 136, female 111, 41.8 years old	Running speed: 2.61–2.69 m/s; Ground running; NA	NA	There is no difference in the location, type, and severity of running-related injuries
Wakeling et al. [9]	Canada	Shaw C41 (soft) Shaw C61 (hard)	6 male 3, female 3 male: 26.0 ± 2.5 years old, female: 23.3 ± 4.1 years old	Running speed: 2.5–4.2 m/s; Ground running; NA	There is no difference in running speed and step length.	NA
Willwacher et al. [10]	Germany	Normal midsole (0.76 ± 0.01 N/mm) Medium hardness midsole (7.11 ± 0.22 N inch mm, 1.9 mm carbon fiberboard) High hardness midsole (16.16 ± 0.20 N inch mm, 3.2 mm carbon fiberboard)	19 male 25.3 ± 2.2 years old	Running speed: 3.5 ± 5% m/s; Ground running; Heel strike	Medium-high overall support period time & pedal time ↑ (ordinary) High metatarsophalangeal joint energy negative function ↓ positive function ↑ (normal-medium) There is no difference between contact time and braking time	NA
Oriwol et al. [17]	Germany	Shaw C52 (M1) Shaw C52 + 36 mm (Shaw C62) (M2) Shaw C52 + 52 mm (Shaw C62) (M3) Shaw C52 + 58 mm (Shaw C62) (M4) Shaw C52 + 79 mm (Shaw C62) (M5) Shaw C52 + 89 mm (Shaw C62) (M6) Shaw C52 + 104 mm (Shaw C62) (M7)	16 male 29.4 ± 6.8 years old	Running speed: 3.5 ± 0.1 m/s; Ground running; Heel strike	There was no difference in all hindfoot motion variables	NA
2. The influence of midsole thickness on running performance/injury						
Chambon et al. [8]	France	Barefoot 0 mm midsole 2 mm midsole 4 mm midsole 8 mm midsole 16 mm midsole	15 males 23.9 ± 3.2 years old	Running speed: 3.3 m/s Ground running; NA	Barefoot and 0 mm midsole stance time ↓ (16 mm midsole)	There is no difference between the peak value of vertical ground reaction force and the peak acceleration of the tibia
Law et al. [12]	China	1 mm midsole 5 mm midsole 9 mm midsole 21 mm midsole 25 mm midsole 29 mm midsole	15 Male 31.4 ± 13.2 years old	Run-on a treadmill and follow the ground after testing the optional speed.	1 mm & 5 mm support interval ↓ (25 mm & 29 mm) There is no difference in landing angle and step length	1 mm & 5 mm Vertical loading rate ↑ (25 mm & 29 mm)
3. The influence of the midsole material/structure on running performance/injury						
Wang et al. [22]	China	EVA midsole (EVA) Polyurethane midsole (PU1) Polyurethane midsole (PU2)	15 male 21.2 ± 1.8 years old	Outdoor ground running; Heel strike	EVA & PU1 peak force after all running distances (PU2) PU1 200–300 km after peak strength (0 km) EVA energy regression (PU1 & PU2)	NA
Wunsch et al. [24]	Austria	Standard foam bottom (foam) Leaf spring midsole (spring)	10 Male 33.1 ± 7.1 years old	Ground running; NA	Spring step length, step frequency, and oxygen consumption ↑ There is no difference in strike pattern	NA
Wunsch et al. [23]	Austria	Standard foam bottom (foam) Leaf spring midsole (spring)	9 Male 32.9 ± 6.1 years old	Running speed: 3.0 ± 0.2 m/s; Heel strike	Spring hip joint energy absorption and ankle energy production ↓ Spring soleus muscle, gastrocnemius muscle strength ↓	NA

Note. NA: not available; LAT: lateral hardness; CON: control group; SS: soft rearfoot/ soft forefoot, soft heel/soft front palm; MM: medium rearfoot/medium forefoot, medium heel/medium front palm; HH: hard rearfoot/hard forefoot, hard heel / hard front palm; SH: soft rearfoot/hard forefoot, soft heel/hard front palm; HS: hard rearfoot/soft forefoot, hard heel/soft front palm, ↑indicates higher while↓indicates lower.

**Table 2 bioengineering-09-00497-t002:** The influence of longitudinal bending stiffness on running performance and related injuries.

Author and Year	Country	Experimental Shoes	Participants	Methods	Results	
					Performance	Injury
Hoogkamer et al. [29]	America	Nike Zoom Vaporfly (NV, Mechanical test deformation 11.9 mm) Nike Zoom Streak 6 (NS, Mechanical test deformation 6.1 mm) Adidas BOOST 2 (AB, Mechanical test deformation 5.9 mm)	18 Male 23.7 ± 3.9 years old	Running speed: 3.89, 4.44 & 5.0 m/s; Ground running	NV Energy loss ↓ (NS & AB)	NA
Madden et al. [27]	Canada	Ordinary running shoes (CON) 185% stiffer running shoes (STI1) 335% stiffer running shoes (STI2)	10 Male NA	Running speed: starts with 2.2 ± 0.2 m/s, increases every two minutes; Indoor track running; Heel strike	There is no difference in the running economy. STI1 & STI2 Peak bending and maximum dorsiflexion rate of MTPJ ↓ (CON)	NA
Oh et al. [28]	South Korea	1.5 Nm/rad Rigid running shoes 10 Nm/rad Rigid running shoes 24.5 Nm/rad Rigid running shoes 32.1 Nm/rad Rigid running shoes 42.1 Nm/rad Rigid running shoes	19 NA 24.7 ± 3.8 years old	Running speed: treadmill runs below the anaerobic threshold; Ground running	The greater the stiffness, the time of support period and pedal extension period ↑ The greater the stiffness, the flexion angle of the MTPJ ↓ The greater the stiffness, the average angular impulse of the MTPJ ↓	NA
Roy et al. [26]	Canada	18 N/mm Rigid running shoes (CON) 38 N/mm Rigid running shoes (STI) 45 N/mm Rigid running shoes (STIEST)	13 NA 27.0 ± 5.1 years old	Treadmill running; Heel strike	STI Maximum oxygen consumption rate ↓ (CON) STI 1 % Energy metabolism ↓ (CON) EST Peak torque of ankle joint ↑ (STI& CON) There was no difference in energy absorption and muscle activation of the MTPJ	NA
Stefanyshyn et al. [13]	Canada	0.04 N·m·deg rigid running shoes (CON) 0.25 N·m·deg Rigid running shoes (MED) 0.38 N·m·deg Rigid running shoes (STI)	5 Male 32.0 ± 13.8 years old	Running speed: 4.0 ± 0.4 m/s; Ground running	STI Energy loss of MTPJ ↓ (MED & CON) There is no difference in energy storage and reuse of MTPJ and energy production and absorption of the hip, knee, and ankle joints of the lower extremities	NA
Stefanyshyn et al. [25]	Canada	Ordinary running shoes (CON) 42 N/mm Rigid running shoes (S42) 90 N/mm Rigid running shoes (S90) 120 N/mm Rigid running shoes (S120)	34 Male 30, Female 4 NA	Running speed: 20 m Sprint, Maximum speed; Ground running	S42, S90 & S120 Sprint time ↓ (CON)	NA
Willwacher et al. [19]	Germany	0.65–0.76 N/mm Rigid running shoes (CON) 5.29–7.11 N/mm Rigid running shoes (MED) 16.16–17.10 N/mm Rigid running shoes (STI)	19 Male 25.3 ± 2.2 years old	Running speed: 3.5 ± 5 % m/s; Ground running	MED & STI Lever arm of the ground reaction force of all joints ↑ (CON) MED Average torque of ankle joint ↓ (CON & STI) STI Negative work of MTPJ ↓, Positive work ↑ (CON & MED) MED & STI Support period and pedal extension period ↑(CON) CON Range of motion and maximum dorsiflexion angle of MTPJ ↑ (MED & STI)	NA

Note. NA: not available; CON: control; STI: stiff; STIEST: stiffest; MTPJ: metatarsophalangeal joint, ↑indicates higher while ↓indicates lower.

**Table 3 bioengineering-09-00497-t003:** The influence of running shoes heel-toe drop on running performance and related injuries.

Author and Year	Country	Experimental Shoes	Participants	Methods	Results	
					Performance	Injury
Besson et al. [35]	France	metacarpal heel difference 0 mm (D0) metacarpal heel difference 6 mm (D6) metacarpal heel difference 10 mm (D10)	14 female 21.4 ± 4.7 years old	Running speed: self-selected; Ground running; Heel strike	D0 ground contact angle, ankle dorsiflexion angle at the beginning of the support period, and the last 40% period ↓ (D6&D10) Before and after the first stage of the D0 support period, the ground reaction force ↑ (D6&D10) D0 pedal and stretch time ↑ Braking time ↓ (D6 & D10) D0 Net moment of ankle flexion during braking ↑ Net moment of ankle flexion during pedal and extension ↓ (D6&D10) There is no difference in the angle of the hip and knee joint and the time of support period.	NA
Chambon et al. [11]	France	metacarpal heel difference 0 mm (D0) metacarpal heel difference 4 mm (D4) metacarpal heel difference 8 mm (D8) barefoot (BF)	12 male 21.8 ± 2.0 years old	Running speed: Self-selected speed; Treadmill & ground running; Heel strike	NA	BF Loading rate of ground reaction force ↑ (D8)
Malisoux et al. [7]	Luxembourg	metacarpal heel difference 0 mm (D0) metacarpal heel difference 6 mm (D6) metacarpal heel difference 10 mm (D10)	553 Male 341, female 212 18–65 years old	Running speed: 2.64 m/s; Outdoor ground running; Heel strike	NA	D6 & D0 Injury risk of occasional runners ↓, injury risk of regular runners ↑ There is no difference in overall damage risk.
Malisoux et al. [34]	Luxembourg	metacarpal heel difference 0 mm (D0) metacarpal heel difference 6 mm (D6) metacarpal heel difference 10 mm (D10)	59 Male 42, female 17 18–65 years old	Running speed: Self-selected; Treadmill and follow the ground	D6 & D10 Adduction angle of the knee joint ↑ D0 There is no difference in support time, flight time, step frequency, step size, and vertical displacement of the hip joint.	NA
De Minds et al. [33]	Belgium	metacarpal heel difference 0 mm (D0) metacarpal heel difference 4 mm (D4) metacarpal heel difference 8 mm (D8) metacarpal heel difference 12 mm (D12)	14 male 27.0 ± 10.0 years old	Ground running; Heel strike	D8 & D12 Maximum pressure center offset in front and rear direction ↑ (D4) D8 Range of pressure center in front and rear direction ↑ (D0) There is no difference in the parameters of the pressure center in the inner and outer directions.	NA
TenBroek et al. [31]	America	metacarpal heel difference 3–3 mm (thin) 9–14 mm (medium) 12–24 mm (thick) barefoot	10 male 18–55 years old	Running speed: 3 m/s; Treadmill running; Heel strike	Barefoot & Angle of ankle dorsiflexion with thin touch to the ground ↓ (medium & thick) Barefoot & Thin Verticality of lower limbs at the moment of touching the ground ↑ (thick) Medium & Thick Flexion and offset angle of the knee joint ↑ (thin & Barefoot) Thickness joint offset angle ↑ (Barefoot) Thin Support period time ↑ (Medium & Thick) Barefoot & thin Peak acceleration of tibia ↑ (Other circumstances) Medium Peak acceleration of tibia ↑ (Thick)	NA
TenBroek et al. [32]	America	The difference between palms and heels 3–3 mm (thin) 9–14 mm (medium) 12–24 mm (thick)	10 male 18–55 years old	Running speed: 3 m/s; Treadmill running; Heel strike	Thin & Medium Angle of metatarsal flexion of ankle joint touching the ground ↑ (thick) Thin Extension angle of touchdown knee joint ↑ (Medium & Thick) Thick Flexion angle of the knee joint in the middle of bracing ↑ (Medium) Thickness joint offset angle ↑ (Thin& Medium) Thick Support period time ↑ (Thin & Medium)	NA

Note. NA: not available; BF: barefoot, ↑indicates higher while ↓indicates lower.

**Table 4 bioengineering-09-00497-t004:** The influence of running shoes’ weight on running performance and related injuries.

Author and Year	Country	Experimental Shoes	Participants	Methods	Results	
					Performance	Injury
Divert et al. [36]	France	Barefoot; Ultra-thin diving socks Ultra-thin diving socks +150 g/piece Ultra-thin diving socks +350 g/piece sports shoes 150 g/piece sports shoes 350 g/piece	12 male 24.0 ± 5.0 years old	Running speed: 3.61 m/s; Treadmill running	Shoe weight increases oxygen consumption ↑	NA
Franz et al. [37]	America	Barefoot Barefoot + 150 g/piece Barefoot + 300 g/piece Barefoot + 450 g/piece Nike Mayfly Nike Mayfly + 150 g/piece Nike Mayfly + 300 g/piece Nike Mayfly + 450 g/piece	14 male 29.8 ± 7.3 years old	Running speed: 3.35 m/s; Treadmill running; Heel strike	There is no difference in oxygen consumption between bare feet and shoes Oxygen consumption increases by 1% for every 100 g shoe weight increase	NA
Hoogkamer et al. [38]	America	Ordinary sports shoes Ordinary sports shoes + 100 g/piece Ordinary sports shoes + 300 g/piece	18 male 24.2 ± 3.3 years old	Running speed: 3.5 m/s; Treadmill running	Oxygen consumption increases by 1.11% for every 100 g weight increase. The running time of ordinary sports shoes is less than 3000 m (+ 100 g/ only & + 300 g/only) For every 100 g weight increase, it increases by 0.78% during 3000 m running.	NA

Note. NA: not available, ↑indicates higher while ↓indicates lower.

**Table 5 bioengineering-09-00497-t005:** The influence of running shoes’ heel flare on running performance and related injuries.

Author and Year	Country	Experimental Shoes	Participants	Methods	Results	
					Performance	Injury
Clarke et al. [39]	Britain	The inclination of both heels 30° The inclination of both heels 15° No heel inclination	10 NA NA	Running speed: 3.8 m/s; Treadmill running; Heel strike	The heel inclination angle reduces the maximum internal rotation angle and the total amount of hindfoot movement ↑ Heel inclination reduces the arrival time of maximum internal rotation velocity ↓	NA
Nigg et al. [40]	Canada	Outside heel inclination 16° No outside heel inclination Round heel	14 male NA	Running speed: 4.0 ± 0.2 m/s; Ground running Heel strike	The heel inclination angle increases the initial internal rotation angle of the foot ↑ There is no difference in total internal rotation angle.	There is no difference in vertical impact force.
Stacoff et al. [41]	Canada	Outside heel inclination 25° No outside heel inclination Round heel	5 male 28.6 ± 4.3 years old	Running speed: 2.5–3.0 m/s; Ground running; Heel strike	No difference in internal and external rotation between the tibia and calcaneus. There is no difference in the speed of hindfoot valgus and maximum valgus.	NA

Note. NA: not available, ↑indicates higher while ↓indicates lower.

**Table 6 bioengineering-09-00497-t006:** The influence of running shoes’ heel stabilizers on running performance and related injuries.

Author and Year	Country	Experimental Shoes	Participants	Methods	Results	
					Performance	Injury
Li et al. [43]	China	No heel cup (N-HC) Heel cup. (HC)	16 Male 10, Female 6 NA	Ground running	Load of plantar fascia and calcaneus after HC4 ↓ (N-HC)	HC heel pain ↓ (N-HC)
Wang et al. [42]	China	Rubber heel cup (Tender-Stride) Rubber heel cup (Tuli’s) Plastic heel cup No heel cup	16 Male 10, Female 6 NA	Running speed: 2.78 m/s; Treadmill running	Plastic heel pad thickness ↑ (rubber) Rubber-plastic heel shock absorber ↑ (no heel cup)	NA
Alcantara et al. [46]	America	Heel stabilizer Heel-less stabilizer	14 Male 9, Female 5 29 ± 17.4 years old	Running speed: 3.35 m/s; Treadmill running; Heel strike	There is no difference in the range of heel motion and tibial horizontal motion.	NA
Ferrandis et al. [45]	Spain	Heel-less stabilizer (P1) Heel stabilizer (P2) Heel stabilizer + vertical sticker (P3) Tighten the heel laces (P4) Tighten the shoelace on the front foot (P5)	10 Male 7, Female 3 NA	Running speed: 3.57 m/s; Treadmill running; Heel strike	Heel stabilizer + vertical foot valgus angle peak ↓ (Other circumstances)	NA
Jorgensen, [44]	Switzerland	Bear foot Heel stabilizer Heel-less stabilizer	11 Male 6, Female 5 25.5 years old	Running speed: 2.5 m/s & 3.1 m/s; Treadmill running	The maximum oxygen uptake of heel stabilizer, tibial acceleration, and the activity of triceps and quadriceps femoris at the moment of landing ↓	NA

Note. NA: not available; NMEC: non-heel cup; HC: heel cup, ↑indicates higher while ↓indicates lower.

## Data Availability

All data generated or analyzed during this study are included in this published article.

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
