# Peer review of "The Implications of Sports Biomechanics Studies on the Research and Development of Running Shoes: A Systematic Review"

_bioengineering, 2022, doi:10.3390/bioengineering9100497_

Round 1
Reviewer 1 Report
General comments
In the study entitled “The implications of sports biomechanics studies on the research and development of running shoes: A systematic review”. 42 articles met the eligibility criteria and included to examine the effect of basic shoe constructions on running biomechanics. The included studies mainly covering the influence of midsole, longitudinal bending stiffness, heel-toe 19 drop, shoe mass, heel flare, and heel stabilizer on running-related performance and injuries. The study demonstrated that running shoe design, target groups tend to influence running performance and injury risk; thickness of 15-20mm, hardness of Asker C50-C55 of the midsole, the design of the medial or lateral heel flares of 15°, the curved carbon plate, and the 3D printed heel cup may beneficial to optimize performance and reduce running-related injuries. After reviewing the manuscript, I do have several suggestions that would further improve the quality of the manuscript.
Introduction
1. The introduction part only present evidence related to midsole, longitudinal bending stiffness, heel-toe drop on running biomechanics, could you add the relative references on shoe mass, heel flare, and heel stabilizer. Which will make your manuscript seems more logical.
Methods
1. Has this manuscript been registered in PROSPERO? Please add the priori established protocol in your manuscript, or explain why you not attached it.
2. Line 82: Five electronic literature databases (Google Scholar, PubMed, ScienceDirect, Scopus, and Web of Science) were used in your study, why you said six databases in your abstract (line 17)?
3. Line 120: Because of the important information has been presented in flow chart, please simplify this paragraph to make your manuscript looks more concise.
4. Line 140: “The overall results were shown in Figure 2”, given that you said the risk of bias in the 42 included studies were assessed, why you only presented 41 studies in Figure 2? Seeing that your eligibility criteria: the research methods must involve the corresponding statistical analysis and offer quantitative results, why you included it rather than remove it, please explain it.
Conclusion
1. Running kinematic and kinetic can be affected by foot strike pattern, what are the running shoes demand differences for runners with different foot strike pattern based on your study parameters?
2. In order to optimize the running performance and reduce running-related injuries, whether it necessary to design new running shoe types for runners with forefoot strike pattern? Could you give some suggestions based on your systematic review?
Author Response
Dear Reviewer 1, Thank you very much for your attention and comments concerning our manuscript. Please see the attachment.

Reviewer 2 Report
Overall, this systematic review has been well-designed and conducted. Moreover, the quality of the scientific and academic writing style is very high.
The review includes a considerable number of studies and it is dense of evidence. Therefore, it could provide an important contribution on this area of research.
Minor suggestions are provided to further improve the quality of manuscript.
Please revise the style of citation in text as the example highlighted in the introduction. Instead of "Nigg (2001)[3],..." write "Nigg [3],..."
Please add a space between data and metric unit in text and tables.
Please avoid the repetition of "included"

Author Response
Dear Reviewer 2, Thank you very much for your attention and comments concerning our manuscript. Please see the attachment.

Round 2
Reviewer 1 Report
Thank you for incorporating reviewers’ comments to improve the manuscript. The authors have adequately addressed my concerns. I am not further questions for the manuscript.